# Over-reassurance and undersupport after a 'false alarm': a systematic review of the impact on subsequent cancer symptom attribution and help seeking

Cristina Renzi, Katriina L Whitaker, Jane Wardle

Department of Epidemiology and Public Health, University College London, Health Behaviour Research Centre, London, UK

**Correspondence to**
Dr Cristina Renzi;
c.renzi@ucl.ac.uk

## ABSTRACT

**Objectives:** This literature review examined research into the impact of a previous 'all-clear' or non-cancer diagnosis following symptomatic presentation ('false alarm') on symptom attribution and delays in help seeking for subsequent possible cancer symptoms.

**Design and setting:** The comprehensive literature review included original research based on quantitative, qualitative and mixed data collection methods. We used a combination of search strategies, including in-depth searches of electronic databases (PubMed, EMBASE, PsychInfo), searching key authors and articles listed as 'related' in PubMed, and reference lists. We performed a narrative synthesis of key themes shared across studies.

**Participants:** The review included studies published after 1990 and before February 2014 reporting information on adult patients having experienced a false alarm following symptomatic presentation. We excluded false alarms in the context of screening.

**Primary and secondary outcome measures:** We evaluated the effect of a 'false alarm' on symptom attribution and help seeking for new or recurrent possible cancer symptoms.

**Results:** Overall, 1442 papers were screened and 121 retrieved for full-text evaluation. Among them, 19 reported on false alarms and subsequent symptom attribution or help seeking. They used qualitative (n=14), quantitative (n=3) and mixed methods (n=2). Breast (n=7), gynaecological (n=3), colorectal (n=2), testicular (n=2), and head and neck cancers (n=2) were the most studied. Two broad themes emerged underlying delays in help seeking: (1) over-reassurance from the previous 'all-clear' diagnosis leading to subsequent symptoms being interpreted as benign, and (2) unsupportive healthcare experiences in which symptoms were dismissed, leaving patients concerned about appearing hypochondriacal or uncertain about the appropriate next actions. The evidence suggested that the effect of a false alarm can persist for months and even years.

**Conclusions:** In conclusion, over-reassurance and undersupport of patients after a false alarm can undermine help seeking in the case of new or recurrent potential cancer symptoms, highlighting the need for appropriate patient information when investigations rule out cancer.

## Strengths and limitations of this study

- The review addresses an under-researched issue, which impacts on a large number of individuals as more than 80% of patients undergoing cancer investigations receive a 'non-cancer' diagnosis (here termed 'false alarm').
- By integrating the available evidence from qualitative, quantitative and mixed methods studies, this review allowed us to identify areas that need to be addressed in order to reduce the risk of delayed help seeking after a previous false alarm.
- Over-reassurance and undersupport of patients can be an unintended consequence of a false alarm leading to delays in help seeking for subsequent cancer symptoms. The effect on delayed help seeking can persist for months and even years.
- The included studies were mainly based on qualitative data collection methods and were limited by small sample size, retrospective design and lack of control groups.
- Prospective studies are needed to identify the appropriate forms of patient information to avoid unintended consequences of false alarms on subsequent symptom attribution and help seeking.

## INTRODUCTION

Patient, doctor and system delays have all been implicated in poorer cancer survival,[1] [2] with particular concern in the UK that these factors are leading to worse cancer outcomes compared with other countries.[3–7] Public awareness campaigns designed to promote earlier presentation with potential cancer symptoms, alongside improved access to diagnostic investigations, have been increasingly advocated to diagnose cancer at an early stage and improve prognosis.[8] [9] However, only a minority of symptomatic individuals undergoing urgent cancer investigations are diagnosed with cancer, with more than 80% receiving an 'all-clear' or non-cancer diagnosis (here called a 'false alarm').[10–12] This

makes it important to consider the possible unintended consequences of a false alarm.

Several studies[11] [13–15] have shown that investigations for a suspected cancer can have negative effects, even for individuals ultimately diagnosed with a benign condition. Anxiety, psychological distress[11] [14] and immunoendocrine changes[15] can persist for weeks or months after a benign diagnosis. In addition, an association between false alarms and subsequent delayed diagnosis has been reported for various cancers,[2] [6] [16–19] with both patients and healthcare providers contributing to delays.[6] However, evidence on the specific processes linking a false alarm to subsequent delays in help seeking is fragmentary. A qualitative synthesis of patients' help seeking highlighted the influence of a benign diagnosis on subsequent symptom attribution as well as worry about wasting the doctor's time as two important factors.[20] Delay in help seeking has also been attributed to the distress caused by a false alarm, and to reassurance from a benign diagnosis.[21]

Several studies have examined the psychological impact of benign or false-positive results of cancer screening[11] [22] [23] and some broader inferences can be made based on their findings. However, the psychological and behavioural consequences of a screening-related false alarm might not be generalisable to symptomatic patients, as highlighted by previous studies.[24] Thus, for our review, we focused specifically on symptomatic patients.

According to the model of pathways to treatment,[25] the process to diagnosis is dynamic with 'forward and backward movement'. The speed and direction of progress through the diagnostic pathway is influenced by patient, healthcare and disease-related factors. This dynamic process involves both patients and healthcare providers reconsidering and reappraising symptoms repeatedly over time. Following a previous all-clear diagnosis, emotional and cognitive factors might play a role in influencing symptom attribution and help seeking, affecting subsequent progress through the diagnostic pathway.

The aim of this study is to review the available international literature to increase our understanding of the processes linking an all-clear diagnosis to subsequent delays, and in particular to examine the impact on subsequent symptom attribution and help seeking.

## METHODS

The literature review included original research using quantitative, qualitative and mixed data collection methods. Identification of relevant qualitative papers is often difficult because indexing is less well developed than for quantitative studies,[20] so we relied on a combination of search strategies, including in-depth searches of electronic databases (PubMed, EMBASE and PsychInfo) using MeSH and free-text key words, searching names of key authors and articles listed as 'related' in PubMed, and searching the references in relevant publications.

The systematic search combined sets of the following groups of keywords: (1) cancer; (2) delay, diagnostic interval, diagnostic pathway; (3) benign, negative, false alarm, all-clear, non-cancer, false positive; (4) symptoms; (5) help-seeking, attitudes, awareness, anxiety, fear, distress, psychological, reassurance; (6) referral, repeat, investigation, examination and test. Within each group, keywords were combined using 'OR' and different groups were combined using 'AND'. Various combinations were used in an iterative process based on the preliminary information obtained from identified sources. This iterative search and a 'snowball' approach, with one reference leading to others,[26] proved essential because the majority of studies were not directly addressing our research question, but relevant information emerged once the sources were examined in detail. Studies were included if they evaluated the effect of health examinations that did not result in a cancer diagnosis (here called a 'false alarm') on subsequent symptom attribution, help seeking or time to diagnosis, for new or recurrent possible cancer symptoms.

The focus of the review was on symptomatic patients, because the effect of a false alarm might be different if it occurs in the context of screening rather than symptomatic presentation.[24] [27] We therefore excluded studies on false alarms following screening. We also excluded studies examining only the emotional effects of investigations for suspected cancer, as previous reviews are available on this topic.[11] [13–15] Publications on childhood cancers were excluded, as were editorials and reviews. We included studies published after 1990 and before February 2014 and no language restrictions were applied.

Initially, one reviewer (CR) conducted the search and screened titles and abstracts. After having excluded irrelevant studies, two reviewers (CR and KLW) independently evaluated the full text of the remaining publications, appraised the studies and performed data extraction. Any disagreement was resolved via a discussion.

In order to extract relevant data, we followed standard methods:[28] the papers were read systematically by two reviewers (CR and KLW), key concepts were recorded and their relationship with a false alarm was explored. Papers were read repeatedly in order to identify additional concepts and identify common or contrasting themes across studies. Using the extracted results, we developed textual summaries and tables, which enabled us to identify emerging themes. All three reviewers (CR, KLW and JW) examined and discussed the findings of individual studies and by comparing similarities and contrasting findings we condensed the number of themes. The level of agreement between reviewers in identifying key themes was high, with only minor disagreements initially regarding some subthemes that were later collapsed into broader categories. Employing an iterative process with discussions between all three reviewers, a consensus was reached and we developed a final narrative synthesis of key themes shared across studies.[29] We have used relevant quotes from selected qualitative and mixed studies to illustrate our findings.

A systematic evaluation of the quality of the evidence was performed assigning a quality score to each reference according to the Mixed Methods Appraisal Tool (MMAT).[30] The MMAT is a valid quality assessment tool for systematic reviews including qualitative, quantitative and mixed methods studies and evaluates each study based on various criteria specific for the different study designs. The highest possible score is 100% if all criteria are met. Two reviewers (CR and KLW) assigned quality scores independently. The level of agreement between reviewers was high and minor disagreement regarding only a few subscores was resolved by discussion. Considering the limited number of studies and in line with previous publications,[20] [31] we decided not to exclude studies based on the quality scores, but rather to take an inclusive approach aimed at identifying research that could give a relevant contribution.

## RESULTS

We initially identified 1442 articles, of which 121 were selected, based on the title and abstract, for full-text evaluation (figure 1). Of these, 19 articles reported information on symptomatic patients with a false alarm, and considered the impact on symptom attribution, help seeking or diagnostic delay for subsequent potential cancer symptoms.

The most frequently studied cancer was breast cancer (n=7 studies), followed by gynaecological (n=3), colorectal (n=2), testicular (n=2), head and neck (n=2), brain cancer (n=1) and multiple cancer sites (n=2; table 1). The majority of studies were carried out in the UK (n=6) and the USA (n=6).

Most studies used qualitative methods (n=14), with quantitative (n=3) and mixed methods (n=2) less frequently employed. They were predominantly retrospective or cross-sectional, with only three having a prospective design. Sample sizes were mainly small and varied between 6 and 3005 participants (median 45; mean 242). The MMAT score was 100% for six studies, 75% for three, 50–55% for eight and 25% for two studies (table 1). Shortcomings included insufficient consideration/information regarding the selection and the characteristics of study participants and insufficient consideration of the possible effects of bias, confounding and other methodological limitations on the study findings (further details available on request).

The studies provided information on the following potential consequences of a false alarm: delayed help seeking for cancer symptoms (n=17 studies), time to diagnosis/delay (n=15), experience of reassurance (n=15), symptom attribution (n=11), perceptions of having been dismissed by the doctor (n=10), lack of information or communication (n=7) and psychological effects (anxiety, distress, fear; n=4). Despite differences by cancer site, study populations and data collection methods, two broad themes emerged across studies: 'over-reassurance' and 'undersupport'.

### 'Over-reassurance' following a non-cancer diagnosis

One of the main themes emerging across studies was patients explaining delay in help seeking as due to reassurance from a previous benign or non-cancer diagnosis[21] [32–45] (table 2).

**Figure 1** Flow of studies.

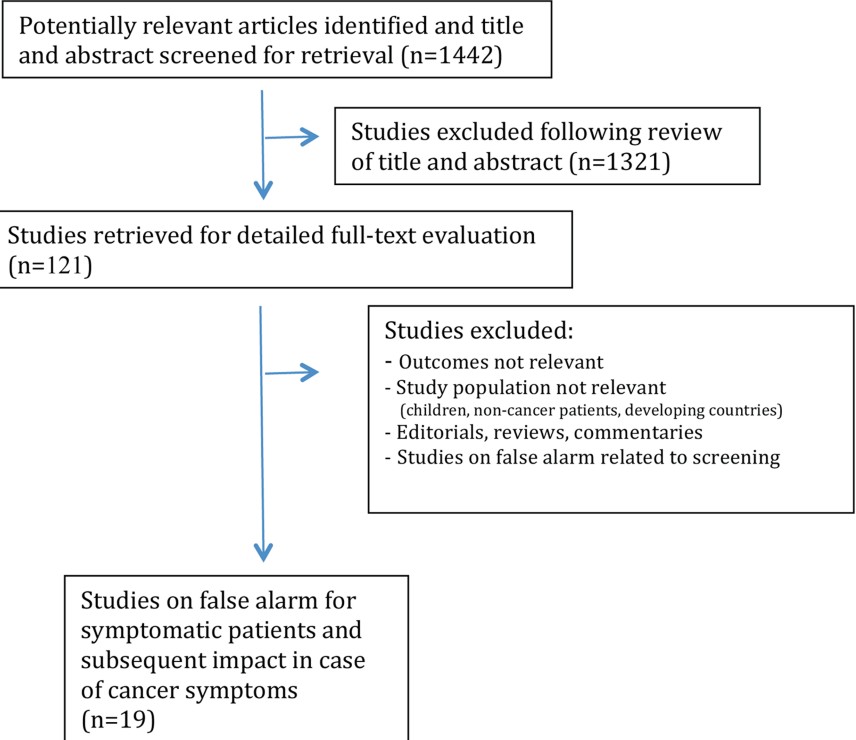

Potentially relevant articles identified and title and abstract screened for retrieval (n=1442)

Studies excluded following review of title and abstract (n=1321)

Studies retrieved for detailed full-text evaluation (n=121)

Studies excluded:
- Outcomes not relevant
- Study population not relevant (children, non-cancer patients, developing countries)
- Editorials, reviews, commentaries
- Studies on false alarm related to screening

Studies on false alarm for symptomatic patients and subsequent impact in case of cancer symptoms (n=19)

**Table 1** Summary information on included studies

| Authors | Country | Study type and data collection method | Participants | Cancer site | MMAT quality score (%) |
|---|---|---|---|---|---|
| Bain et al[33] | UK | Qualitative interviews | 95 patients with cancer | Colorectal | 100 |
| Beacham et al[27] | USA | Prospective observational (telephone questionnaire) | 37 women with benign breast biopsy following self-detected lump, 65 following screening and control group of 76 women without biopsy | Breast | 50 |
| Chapple et al[35] | UK | Qualitative interviews | 45 patients with cancer | Testicular | 75 |
| Evans et al[38] | UK | Qualitative interviews | 43 patients with cancer | Gynaecological | 100 |
| Facione and Dodd[36] | USA | Qualitative interviews | 39 patients with cancer | Breast | 50 |
| Facione NC and Facione PA[37] | USA | Qualitative interviews | 28 women with possible breast cancer symptoms | Breast | 100 |
| Fitch et al[39] | Canada | Qualitative interviews | 18 patients with cancer | Gynaecological | 25 |
| Gascoigne et al[40] | UK | Qualitative interviews | 6 patients with cancer (and 5 relatives) | Testicular | 75 |
| Granek and Fergus [46] | Canada | Qualitative interviews | 14 patients with cancer (and 7 partners) | Breast | 100 |
| Heisey et al[21] | Canada | Qualitative interviews | 14 patients with cancer; 10 GPs | Breast | 50 |
| Janz et al[24] | USA | Prospective observational (telephone questionnaire) | 83 women with benign biopsy after self-discovered breast problem (and control group of 393 women with no breast problem) | Breast | 75 |
| Jones et al[41] | Australia | Cross-sectional (telephone questionnaire) | 3005 participants from the general population with potential breast cancer symptoms | Breast | 50 |
| Salander et al[42] | Sweden | Qualitative interviews | 28 patients with cancer and 27 spouses | Brain | 50 |
| Scott et al[43] | UK | Qualitative interviews | 57 patients with cancer | Head and neck | 100 |
| Siminoff et al[44] | USA | Qualitative interviews (and review of medical records) | 242 patients with cancer | Colorectal | 100 |
| Tarling et al[47] | UK | Prospective observational (mixed-methods: questionnaire and focus groups) | 55 women with non-cancer diagnosis after urgent referral for postmenopausal bleeding (35 completed questionnaire and 15 completed focus groups) | Gynaecological | 55 |
| Tishelman et al[34] | Sweden | Qualitative interviews | 46 patients with cancer (and 29 relatives) | Multiple sites | 50 |
| Tromp et al[32] | NL | Case-series (mixed methods: questionnaire, interviews and physician questionnaire) | 306 patients with cancer | Head and neck | 55 |
| Underwood et al[45] | USA | Qualitative interviews | 46 patients with cancer | Multiple sites | 25 |

GP, general practitioner; MMAT, Mixed Methods Appraisal Tool; NL, the Netherlands.

**Table 2** Factors influencing delayed help seeking in relation to a previous all-clear diagnosis, based on the available evidence

| Main themes | References |
| --- | --- |
| Reassurance | Bain et al;[33] Chapple et al;[35] Evans et al;[38] Facione and Dodd;[36] Facione NC and Facione PA;[37] Fitch et al;[39] Gascoigne et al;[40] Heisey et al;[21] Jones et al;[41] Salander et al;[42] Scott et al;[43] Siminoff et al;[44] Tishelman et al;[34] Tromp et al;[32] Underwood et al[45] |
| Symptom attribution | Bain et al;[33] Beacham et al;[27] Evans et al;[38] Facione and Dodd;[36] Granek and Fergus;[46] Heisey et al;[21] Janz et al;[24] Jones et al;[41] Salander et al;[42] Scott et al;[43] Tromp et al[32] |
| Perception of having been dismissed | Evans et al;[38] Facione NC and Facione PA;[37] Fitch;[39] Gascoigne et al;[40] Granek and Fergus;[46] Heisey et al;[21] Salander et al;[42] Siminoff et al;[44] Tishelman et al;[34] Underwood et al[45] |
| Lack of information/ communication | Evans et al;[38] Facione NC and Facione PA;[37] Fitch et al;[39] Siminoff et al;[44] Tarling et al;[47] Tromp et al;[32] Underwood et al[45] |
| Anxiety, distress, fear | Beacham et al;[27] Chapple et al;[35] Tarling et al;[47] Tromp et al[32] |

Among a general population sample of 3005 women with breast symptoms, the single most common reason for not seeing a doctor when experiencing a breast lump was that they had seen a doctor about it before (reported by 15.7% of women).[41] A study of 242 patients with colorectal cancer (CRC) showed that a delayed diagnosis was associated with having received an initial non-cancer diagnosis (10.2 vs 2.4 months; p<0.001), or having been initially told not to worry or to continue to monitor symptoms (6.8 vs 4.4 months; p=0.006).[44] A study of 306 patients with head and neck cancer showed that delayed help seeking was associated with patients interpreting symptoms as innocent partly because of reassurance during the first visit.[32]

Qualitative studies illustrate how over-reassurance can lead to normalisation of symptoms and subsequent delayed help seeking: "He [physician] gave me an examination and said 'there was nothing there'. So you go home and live with the problem."(No.20; patient with CRC).[33] "He (surgeon) did a colonoscopy…He was relieved because he didn't find anything- so I did nothing for about two years and the blood wasn't getting any worse on the toilet paper."(No. 48; patient with CRC).[33]

Having been reassured by previous examinations, some patients—and some physicians—attributed subsequent symptoms to benign conditions.[21 32 33 36 38 41–43 46] Patients with breast cancer reported retrospectively: "It was fibro [something]. Yeah, benign…So…I don't know…I mean it was a few years later, the breast started to feel a similar kind of way. I said, oh, it's the same thing, you know…And I ignored it."(P9).[21] "When I did an exam one day I found a different lump in each breast. I was used to them…I was thinking, 'Oh, here we go again.' And maybe for a minute you might think 'Gee, I hope it's not positive' but that went very quickly. The more biopsies I had, the less concerned I was that they would be positive." (woman with four prior benign incisional biopsies).[36] Some studies have even shown a decrease in breast self-examination after a benign breast biopsy performed for a self-detected lump.[24 27]

Similarly, patients with ovarian cancer[38] and brain tumour[42] reported delays in help seeking and specialist referral because symptoms were attributed to a previous benign condition or other reasonable explanations after an initial non-cancer diagnosis and negative tests. Likewise, for testicular cancer, a diagnosis such as a cyst or urinary infection led to subsequent interpretation of symptoms by patients and physicians in line with the previous benign diagnosis, with delays in help seeking and diagnosis of up to 12 months.[40]

It is possible that symptom characteristics might moderate the effect of a false alarm.[32–34 38 39 42–44] For example, among women with breast symptoms, a previous visit with a non-cancer diagnosis was a relatively frequent explanation for delayed help seeking in the case of a breast lump (reported by 15.7% of women), while it was less frequently mentioned as a reason for delay in the case of other symptoms, such as swelling in the armpit, pain or change in breast shape or size.[41]

### 'Undersupport' following a non-cancer diagnosis

The second broad theme was the patient's perception of previously not having been taken seriously, or symptoms having been dismissed as unimportant, as well as a sense of humiliation or concern about appearing hypochondriacal[21 34 36–40 42 44–46] (table 2). Women with breast symptoms who had delayed seeking help for a year reported that the delay was influenced by concerns about appearing hypochondriacal or foolish, following an experience of being dismissed or treated with disrespect: "I've had my symptoms dismissed as frivolous twice."[37] "Well, because I'd had identical symptoms over 20 years before, years before, and it had been mastitis. And at that time I had worried about cancer and was basically kind of laughed at and…and I felt foolish about how I'd been so worried…I was very humiliated, I was very embarrassed." (Donna, 63, breast cancer).[46] "…So having been dismissed the first time, I said, I'm overreacting, just leave it alone. So that's why I'm saying that my first experience kind of influenced me even getting the follow-up the first time I noticed any slight change." (P9).[21] Patients with testicular cancer also described long delays before seeking help again for persistent symptoms: "Saw the general (senior?) registrar

who examined me and told me there was nothing wrong with me and gave me one hell of a telling off for not listening to his registrar, and politely told me to bugger off and not to waste his time again."[40] Similarly, patients with gastrointestinal cancer reported: "They had a very negative attitude; I wasn't really believed. They said it was psychosomatic. I was reluctant to try and get help after that."[34]

A number of studies reported that previous visits with a non-cancer diagnosis left the patients frustrated, with a sense that doctors could not help them and uncertainty about what to do next; these factors contributed to subsequent delays.[32 37–39 44 45 47] Of 155 patients with head and neck cancer having initially received an all-clear, 50% waited more than 3 weeks before returning to the doctor, and 10% more than 4 months; some explained their delay with the fact that the doctor could not help the first time.[32] Likewise, lack of explanation about the possible causes and meaning of symptoms, and lack of advice on further actions after investigations for postmenopausal bleeding can delay subsequent help seeking.[47] Women reported a sense of frustration and not knowing what to do in case of recurrent symptoms: "It's the not knowing. It's more frustrating. Why is it still happening? Mine has not changed that much. After going through all that. It puts you off going again, because they don't know and they don't tell you anything else." (P2).[47] Similar explanations for delayed help seeking were reported by patients with breast cancer:[37 45] "I've had no relief from seeing a physician."[37] Also patients with ovarian cancer explained delays as due to frustration and not having previously discussed with the doctor any alternative diagnostic hypothesis or planned any follow-up or further actions.[38] Among patients with CRC, lack of communication of the next steps during the initial visit was associated with longer diagnostic delay (8.2 vs 3.4 months; p<0.001).[44]

The other explanation for delay was anxiety or distress following the previous non-cancer diagnosis, reported by some women with recurrent postmenopausal bleeding after a false alarm,[47] and by some patients with head and neck cancer.[32] Among patients with testicular cancer, some reported fear of painful investigations following previous health examinations: "And then when I did go and see this GP, there was a locum, and he gave me an inspection, and I found it quite uncomfortable, the way he went about the inspection. And so I further delayed. You know, he had referred me to somebody else. And I delayed that…" "…it was excruciatingly painful, you know, I didn't like that, you know, and I suppose anyone does like that sort of thing. Anyway, it was my own fault that I delayed the thing."(T45).[35]

## DISCUSSION

An 'all-clear' or non-cancer diagnosis can be associated with subsequent delays in help seeking in the case of new or recurrent possible cancer symptoms. Our review of a largely qualitative literature has shown that across different cancer sites and study populations, some common themes emerged to help explain the relationship between a false alarm and subsequent delays. The two main themes were 'over-reassurance', resulting in subsequent attribution of symptoms to the initial benign diagnosis or normalising of symptoms, and 'undersupport', resulting in symptomatic patients being unwilling to seek medical attention again. Many of the studies report on prolonged delays, suggesting that the effects of a false alarm can be long-lasting, and may generalise beyond recurrence of the original symptom to new symptoms appearing some time later. In the case of breast symptoms, a benign diagnosis appeared to give some women a false sense of security persisting for many years.

This sense of security is at odds with the need to remain vigilant, particularly in the light of recent evidence showing that women with a histologically proven benign breast biopsy can have a 2–3-fold increased risk of being subsequently diagnosed with breast cancer.[48 49] Around 2% of women with a benign breast disease are diagnosed with breast cancer during the following 7 years,[49] with the lesion being considered a marker for increased risk, rather than a premalignant lesion in itself. In a single institution study in the USA, 13% of breast cancer diagnoses involved women presenting with a palpable mass who had a negative mammogram within the last year, and 21% had had a mammogram 1 year or more before.[50] Also for CRC, there is evidence supporting the need to remain vigilant even after negative investigations: up to 8% of cases are diagnosed within 3–5 years of a negative colonoscopy, possibly because of missed cancers or cancers arising from missed or incompletely removed polyps.[51] In a single institution study in the UK, the diagnostic yield of a second urgent referral, although lower than the first referral (5% vs 10%), is not insignificant.[52]

Our review has shown that undersupporting patients receiving an all-clear diagnosis can negatively impact future symptom interpretation and help seeking. The perception that symptoms were previously dismissed as unimportant was a relevant theme explaining subsequent delays, most often because of not wanting to appear hypochondriacal. Patients' concerns about wasting the doctor's time, which previous studies reported as a common barrier for help seeking in the UK,[20 53 54] was mentioned by some patients, but appeared to play a less relevant role.

Fear of cancer or of the consequences of treatment has been previously shown to be a barrier for help seeking,[20 55] Our review suggested that fear of examinations or high anxiety levels after a false alarm contributed to delays only in a minority of cases. Other factors seemed more relevant, such as a sense of frustration, uncertainty about what to do next and not having discussed any alternative diagnostic hypothesis or follow-up at the time of the initial consultation.

The need to provide patients with more information in the case of a non-cancer diagnosis has also been highlighted in a study on 'straight to test' endoscopy services for suspected CRC:[56] more than 30% of patients would prefer to see a specialist even after normal or benign test results. A clinical encounter providing information before and after diagnostic investigations may be valuable to ensure that bodily sensations are not dismissed following negative examinations, and to discuss next steps in the case of recurrent or new symptoms.

Our review and previous studies[42 57] have shown that over-reassurance from normal test results or a benign diagnosis can influence patients and healthcare providers, possibly affecting time to diagnosis. Planned follow-up soon after the initial diagnosis can help mitigate the risks associated with overconfidence in the first diagnosis; it allows the clinician to apply more conscious problem solving and for the possibility of alternative diagnostic hypothesis to emerge, with symptom changes guiding this process.[58] Primary care physicians can also be undersupported in terms of not having sufficient access to diagnostic investigations.[8 59] For example, 1 in 10 general practitioners in the UK had tests for ovarian cancer refused.[60] Further studies based on healthcare providers' experiences are needed.

In the UK, urgent cancer examinations have risen over time, but this is inevitably followed by a decrease in the diagnostic yield.[52 61] More patients will experience a false alarm as a consequence of initiatives promoting earlier symptomatic presentation and improved access to diagnostic investigations.[8 9] Despite being unavoidable if early diagnosis and survival are to be improved, especially for cancers presenting with non-specific symptoms and in the absence of accurate markers for discriminating between high-risk and low-risk individuals, effort is required to minimise unintended consequences. Significant event audits in primary care have highlighted the need to find a balance between avoiding unnecessary anxiety in symptomatic patients and the potential risks of over-reassuring patients with an all-clear diagnosis.[57] Recommendations similar to those developed for children with acute diseases have been suggested for safety netting and preventing delays in cancer diagnosis.[57 62] These include communicating to patients that there is uncertainty and that more visits might be necessary for reaching a diagnosis, explaining exactly what symptoms merit special attention, giving advice on how to seek help if necessary, and explaining the expected development of the illness over time.

Providing balanced information and involving patients in monitoring their symptoms and bodily sensations are also relevant in other contexts, such as cancer screening,[63] fast-track referral systems[64] and early detection of recurrent cancers.[65] Electronic tools have been developed supporting people with cancer to prospectively collect patient-reported data and for helping clinicians to monitor trends of symptom severity.[66] Similar instruments could also be used to help monitor the evolution of symptoms in individuals with persistent or recurrent symptoms after a false alarm with potential beneficial effects in terms of providing support and limiting over-reassurance.

Even though the relevance of patient self-monitoring and awareness of bodily changes is recognised in cancer awareness campaigns (http://www.cheekycheckup.com.au; http://www.cancerresearchuk.org) and during clinical encounters, there is a lack of specific advice and tools for patients with false alarms.

Our findings on symptomatic patients with a false alarm are in line with some screening-related studies: women with previous negative screening mammograms and later diagnosed with interval breast cancer explained delayed help seeking in part due to previous over-reassurance and undersupport.[67] Moreover, recent systematic reviews on the impact of false-positive screening mammograms in the UK[23] have shown long-lasting distress for up to 3 years and a lower likelihood to reattend subsequent screening assessments. There is some weak indication that these negative effects could be overcome by improving communication and providing tailored information.[68] However, other reviews on false-positive screening results referring to European, Canadian and US populations showed conflicting evidence[22 69] and more research is needed to understand the effects of false alarms following screening as well as following symptomatic presentation.

It should be noted that even though studies referring to screened and to symptomatic individuals can complement each other in the attempt to increase our understanding of the psychological and behavioural consequences of a false alarm, the results are not directly transferable to different contexts. This can be exemplified by studies showing that breast self-examination was more likely to decrease among women with a benign diagnosis following a self-identified lump, while it more likely increased if the breast problem was discovered by the healthcare system.[24]

There are some limitations to our review. The majority of studies did not have the specific objective of evaluating false alarms, and relevant information emerged only after in-depth examination of full-text publications. Thus, we cannot exclude the possibility that some studies were not identified in our review. The included studies were limited by small sample size, retrospective design and lack of control groups. As the majority of studies were retrospective or cross-sectional and based on reports by patients with cancer, recall bias might have influenced the findings. When patients are asked to recall experiences and reasons for delays after having been diagnosed with cancer, their answers might mask a sense of guilt if they neglected symptoms or delayed help seeking.[67] More prospective studies are needed, also including information provided by healthcare professionals.

The majority of studies were conducted in English-speaking countries, mainly the UK and the USA,

with a few from Northern Europe. This might reflect policies and initiatives addressing earlier cancer diagnosis having taken place in these countries. Publication bias might also influence the number of studies from different countries. More international comparisons, including central and southern European countries could provide a different perspective on common issues.

An 'all-clear' diagnosis in terms of cancer can result from a variety of different clinical scenarios, including a true benign diagnosis, a false-negative result or the healthcare provider attributing symptoms to alternative explanations. Our study was not able to stratify by these factors, but we did not identify any specific differences regarding the effect on reassurance, symptom interpretation and help seeking, of either type of, or time since, the false alarm. Larger prospective studies are needed to explore these issues.

In conclusion, we found that a false alarm can influence subsequent symptom attribution and help seeking, principally through patients being either 'over-reassured' or 'undersupported' in relation to future symptoms. Providing patients with appropriate balanced information when investigations rule out cancer may help to prevent subsequent delays. Prospective studies are needed to identify forms of patient information that limit unintended consequences of false alarms.

**Contributors** CR and JW designed the study. CR was responsible for data collection. CR and KLW performed data extraction and appraising studies. CR, KLW and JW contributed to data analysis and interpretation. CR wrote the first draft of the manuscript. All authors reviewed the manuscript and approved the final version. CR is responsible for the overall content as the corresponding author.

**Funding** This work was supported by Cancer Research UK (C48748/A16867).

**Competing interests** None.

**Provenance and peer review** Not commissioned; externally peer reviewed.

**Data sharing statement** Extra data on the quality appraisal of the included studies is available by emailing c.renzi@ucl.ac.uk

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
