## [Reviewer comments · BMJ Open]

ARTICLE DETAILS

TITLE (PROVISIONAL)	Over-reassurance and under-support after a 'false alarm': a systematic review of the impact on subsequent cancer symptom attribution and help-seeking
AUTHORS	Renzi, Cristina; Whitaker, Katriina; Wardle, Jane

VERSION 1 - REVIEW

REVIEWER	Nur Aishah Taib Consultant breast surgeon, University of Malaya, Kuala Lumpur, Malaysia
REVIEW RETURNED	10-Dec-2014

GENERAL COMMENTS	The review was focused on cancer patients with new symptoms after a previous false alarm. The authors reviewed articles with qualitative, quantitative and mixed methods. The authors clearly stated the limitations of this review as they did not use an appraisal check-list as to be inclusive of qualitative papers with the related research question.
--

REVIEWER	Kate Brain Cardiff University School of Medicine Wales
REVIEW RETURNED	11-Dec-2014

GENERAL COMMENTS	This paper reports the findings of a literature review and narrative synthesis regarding the effects of a 'false alarm' symptom on subsequent cancer symptom attribution and symptom presentation behaviour. The paper is well-written but is not based on systematic review/critical appraisal methods, which limits its contribution to the field. However, the findings regarding the influences of individual and healthcare system factors on cancer early diagnosis are potentially interesting and relevant to NAEDI policy, with included studies suggesting that benign diagnoses may lead to false reassurance, normalisation and prolonged help-seeking for new or recurring symptoms. Based on the novelty and relevance of the research question, I recommend re-submission following revisions as suggested below. Abstract This is a fairly minor criticism but the authors refer variously to symptom 'awareness', 'attribution' and 'appraisal', whereas I think it's better to identify one or two conceptual terms and use them consistently (here and throughout the paper).
---

	Introduction The problem of over-reassurance in the face of a false alarm symptom appears to be an important public health issue. The authors cite the statistic that some 80% of individuals who are undergoing cancer diagnostic tests eventually receive an all-clear result, but it would be helpful to clarify whether this statistic relates specifically to referrals based on symptom presentation. Related to the last point, much of the cited background literature seems to refer to the psychological impact of false positive results in cancer screening. Although we can extrapolate from screening studies, they are not directly relevant to the symptom presentation context (as the authors later acknowledge in their Methods). I think this point should be addressed explicitly in the Introduction. Methods Lines 10-17: The reasons for not carrying out a critical appraisal of included studies are not clear to me. The review could be strengthened by a systematic evaluation of the quality of evidence (in several domains e.g. study design, participants, measures), even from a relatively small number of studies. I think this is a deficiency that should be addressed. The authors should also report the level of agreement between reviewers identifying key themes from the included studies. Discussion The discussion would be strengthened by a more detailed critique of the included studies, in particular the limitations associated with retrospective recall bias in cancer patients, who formed the majority of participants in the 19 studies included in this review.
--	---

REVIEWER	Joan Prades Catalonian Cancer Plan, Department of Health, Government of Catalonia, Spain
REVIEW RETURNED	12-Dec-2014

GENERAL COMMENTS	This is an article that deals with a highly relevant topic in cancer care, namely the psychological complexity behind the detection of a suspicion of cancer. It is interesting that the paper considers the perspective of both patients and health care professionals. The authors should be commended for attempting to gather evidence on this very difficult area. The paper is essentially restricted to papers using qualitative methodology. This is an appropriate approach as regard to the issue to deal with, although it makes complex to combine the evidence from the different papers included in the review. The research has nonetheless a clear focus. Some points deserve consideration by the authors, specifically in the discussion section. First of all, the discussion of the evidence should include false alarms in the context of screenings programs and fast-track referral systems (e.g. the 'two-week rule'). Despite these were excluded in the research question, the work done in this field deserved to be mentioned. As a matter of fact, individuals invited to screening programs are not symptomatic, yet the process of involvement take into account information and communication conditions and targets, for instance, in the case of patients with low
---

	literacy skills (Austoker J et al, Endoscopy, 2012). Moreover, fast-track systems to refer patients showing cancer symptoms make a difference in how patients and GPs deal with uncertainty arising from a cancer suspicion. As regard cancer recurrences, Montgomery et al (BJC, 2007), for instance, showed how the majority of them were picked up through symptoms detected by breast cancer patients themselves between scheduled visits. Thus, the need for balanced information and good communication skills in the phase of detection is not a new issue in cancer care, and broadening the focus to make evident such a fact would be advisable. Also, it would be interesting to mention the potential role of patients as observers or monitors of their own symptoms. In this line, when considering the need for achieving balanced information between avoiding unnecessary anxiety in symptomatic patients and the potential risks of over-reassuring patients, no mention is made of 2.0 tools. These could be useful in supporting the task of health care professionals to this end. As an example, the "Cheeky check-up" program in Australia explains in a smooth way breast changes, detailing three specific steps for self-monitoring, and recommending to report any unusual change to the doctor (http://www.cheekycheckup.com.au/).
--	--

REVIEWER	Dr Alex Dregan King's College London
REVIEW RETURNED	12-Dec-2014

GENERAL COMMENTS	The authors conducted a systematic review about the consequences of false-alarm with regards to cancer symptoms. The manuscript is well written and could of general interest. I have only some minor comments that the authors may wish to consider.  1. It's not clear whether the authors used other terms , ie false-positives in their search strategy. There are several recent reviews in the area that need at least mentioning (eg Bond et al., 2013; Lidbirk et al., 1996) 2. The authors may also discuss about the implications of the study in terms of the evidence about the strength of the correlation between self-reports and actual behaviour 3. May also wish to discuss in the Limitations about potential biases associated with retrospective/cross-sectional data 4. Studies heterogeneity? - could exclude the quants papers - very small number for any definitive conclusions 5. May help to discuss the study findings in relation to studies on consequences of false-positives in cancer screening.
--

VERSION 1 – AUTHOR RESPONSE

Reviewer Dr Kate Brain

Abstract

Following the reviewer's suggestion we have consistently used the term 'symptom attribution' throughout the paper and have avoided using other terms such as symptom 'awareness'.

Introduction

We have specified that the cited statistic of 80% of individuals undergoing cancer diagnostic tests and receiving an all-clear result refers to symptomatic patients.

As suggested, we have mentioned in the introduction that even though several studies have examined the psychological impact of benign or false positive results of cancer screening, the psychological and behavioural consequences of a screening-related false alarm might not be generalizable to symptomatic patients. We have therefore focused our review specifically on symptomatic patients. This issue is discussed in more details in the discussion section, and is also in line with the other reviewer's comments.

Methods

We have modified the methods section according to the reviewer's suggestions (see also answer to the editor).

Discussion

We have added a more detailed discussion of the limitations of the included studies, in particular regarding the issue of the retrospective design of the majority of studies and possible recall bias.

Reviewer Dr Joan Prades

We have revised the discussion following the reviewer's suggestion. In particular we have broadened the focus referring also to false alarms in the context of screening. Moreover, as suggested, we have highlighted that the need for balanced information and for involving patients in monitoring their symptoms are also relevant in other contexts, such as cancer screening, fast-track referral systems and early detection of recurrent cancers. In the discussion we have also mentioned the possible development and use of tools to prospectively collect patient reported data to monitor trends of symptom severity (PCM v2.0). In the context of patients monitoring their symptoms we have also mentioned the "Cheeky check-up" program in Australia.

Reviewer Dr Alex Dregan

We are grateful to the reviewer for highlighting the issue regarding the term 'false-positives'. We had included it in our search strategy, but had erroneously omitted to specify this in the methods section. This has now been corrected (see also answer to Editor). We are also grateful to the reviewer for the useful additional references.

As suggested, we have discussed in more details the limitations of the included studies and the possible recall bias due to the retrospective/cross-sectional study design of the majority of studies. Despite the small number of quantitative papers on false alarms, we believe their inclusion in the review provides a different perspective and can be useful in complementing the qualitative and mixed methods paper.

As suggested we have discussed the study findings also in relation to studies on the consequences of false-positive results following cancer screening.